# Business Leadership from a Gender Perspective and Its Impact on the Work Environment and Employee’s Well-Being in Companies in the Basque Country

**DOI:** 10.3390/ijerph19010030

**Published:** 2021-12-21

**Authors:** Antonia Moreno, Fernando Díez, Luana Ferreira

**Affiliations:** Faculty of Education and Sport, University of Deusto, Avda. Universidades 24, 48007 Bilbao, Spain; fdiez@deusto.es (F.D.); luana.ferreira@deusto.es (L.F.)

**Keywords:** leadership, organizational culture, work environment, occupational health, well-being

## Abstract

If the workplace environment is good, the health and well-being of employees will be good too. This research aims to distinguish whether there are differences when it comes to being directed by a man or a woman and whether this affects employees. An ad hoc questionnaire was applied, collecting personal information and including the MLQ-6 S. It was sent by mail and answered by 549 employees of 16 companies in the Basque Country, Spain. A total of 277 (50.5%) men and 272 (49.5%) women participated, among whom there were 63 managers. The methodology shows a double perspective of how employees understand and perceive the differences between male and female business leaders and how managers see themselves exercising this leadership. No significant differences have been perceived. Both men and women believe they make their employees feel good about exercising leadership (M = 42.11%, W = 48.00%) quite often. Employed women consider it more challenging to become leaders and reconcile their work-life. Men do not think so. Communication is the tool that women managers know best how to handle and where men seem to fail more. Working on it could achieve more business success and better health in employees.

## 1. Introduction

A commonly held belief among managers is the notion that healthier employees contribute more to the effectiveness of their organizations [1]. Unfortunately, managers are often ignorant of the consequences of their actions on the well-being of subordinates [2] and the associated organizational consequences [3].

In the occupational health research field, job dissatisfaction (i.e., an unpleasant emotion when one’s work is frustrating and blocking the affirmation of their values) [4] has been attracting attention as a predictor of sickness absence, as well as of poor mental health (i.e., anxiety, burnout, depression, and low self-esteem) and physical health (i.e., cardiovascular disease and musculoskeletal disorders) [5]. Depressive symptoms in the workplace can result in substantial adverse effects, including impaired job performance [6], extended illness-related absences [7], and increased financial costs associated with disability pensions [8].

Well-being and mental health are affected by job strain and work characteristics such as lower perceived job control, increased job demands, and lower social support by supervisors (i.e., emotional concern, instrumental and information support, and appraisal). The relationship can be bidirectional, so as well-being and mental health can affect work, work characteristics can have a more significant adverse effect on well-being than vice-versa [9].

Many companies and health insurance funds have recognized the potential of changes in the working environment to contribute to maintaining employees’ health. Beyond the legally required occupational safety measures, companies are increasingly offering additional steps to promote health for their employees [10]. All joint actions taken by employers, employees, and society to improve health and well-being at work are called Workplace Health Promotion (WHP) [11].

The role of workplace environments in the prevention and control of non-communicable diseases (NCDs) has been emphasized by the international community since 2012/2013, when the United Nations and World Health Organization (WHO) called for a ‘whole-of-government, whole of-society’ approach to NCDs, including the creation of an ‘enabling environment for healthy behaviors among workers’ [12].

### 1.1. The Impact of Leadership on Workers’ Health and Well-Being

The occurrence of workplace conflict is unavoidable as it is a part of everyday business life; however, it is manageable [13,14]. Workplace health promotion (WHP) includes the combined efforts of employers, employees, and society [15]. Workplace health programs have demonstrated substantial benefits for businesses, such as decreased absenteeism [16,17,18], increased likelihood of smoking cessation, and decreased systolic blood pressure in employees.

There is evidence that employees’ multiple and varied experiences in the workplace impact their occupational health [19], and many of these experiences depend on how they are managed and supervised [3]. Some researchers in occupational health psychology argue that leadership should be considered an intervention area that can improve employee well-being [20,21].

In this context, an increasingly important aspect of leadership encompasses bosses’ empowering leadership behaviors towards their subordinates [22]. Bosses can delegate more power to their subordinates, increase their responsibilities, encourage independent decision-making, share information and knowledge with them, develop their skills, or encourage them to take risks or propose new ideas [23]. However, supervisors can also behave in the opposite manner, what would affect their employees’ health and well-being at work [24].

Managers play an essential role in facilitating new practices by making a process easy or more manageable or by promoting and helping the process forward [25]. Their experience of the implementation process in health care and how it affects the implementation outcome are still largely unknown [26,27,28,29].

A workplace is a place where various relationships are established. The leader is among those who play an important role in creating (non-)functional relationships in the workplace; they present a distinctive personality in the workplace, which manifests according to a specific type of behavior. Employees emotionally perceive the leader’s behavior, and such behavior should therefore be helpful and not harmful to them. The inevitable interactions and (co-) influences that occur in the workplace may positively or negatively affect employees, the results of which may be perceived through the lens of sickness absence [30]. Bernstrom and Kjekshus [31] (2012) define sickness absence as all absences due to ill health. Feeling good at work is a strategic issue for all partners in a company [32], as the well-being concept refers not only to the absence of pathologies or physical and mental illnesses and disabilities but also to a more general feeling of satisfaction and fulfillment in professional and extra-professional life.

This key element of human resources has become an essential element of any thoughtful and global managerial policy, clearly identified as a potential lever for the growth and improvement of a company’s financial results [33].

The International Labor Organization defines psychosocial risk at work as interactions between the individual and a range of workplace factors, including job design, management, and the organizational environment that can have a hazardous influence on an employee’s health [34]. There is increasing interest at the governmental level in reducing workplace absenteeism and work disability due to adverse (psychosocial) working conditions [35]. However, only 30% of European companies have implemented measures dealing with psychosocial risks within an organization-centered management approach [36].

### 1.2. Work Environment and Organizational Culture

Work provides a sense of identity, love and belongingness, and a social outlet [37,38]. Psychologically, work serves many dimensions of psychological and mental health/well-being. It increases self-esteem, [37] confidence, gives people a sense of purpose, self-worth, autonomy, competence, independence, achievement, and overall improves the quality of life, satisfaction, and accomplishment [39] of people, strengthening their sense of self-integrity. Furthermore, workplace social relations help to control and buffer the stress people experience in life. As such, the notion of work environment goes beyond the physical setup where employees perform their everyday tasks and also comprehends the net of relationships that are formed at both horizontal and vertical hierarchical levels [40].

A healthy work environment has a high value for organizations and their co-workers, and the quality and style of leadership influence the level of health [41]. Research has shown that satisfaction with the work environment has a substantial positive impact on employee health across countries [42]. There is a connection between the organizational environment, organizational efficiency, work-related stress, and co-worker mental health [43].

On the other hand, work stressors (such as work pressure, workload, organizational constraints, work hours, etc.) demonstrated to have plain to strong correlations with psychological and physical well-being. However, the threshold for affective reactions (for example, anxiety and emotional exhaustion) is lower than for physical symptoms (headache, fatigue, gastrointestinal issues) [44]. The work environment is also one of the two most important determinants of workplace bullying [45,46], which is related to several health problems [47,48]. In this regard, research has demonstrated that ethical leadership can contribute to an improved work environment and to decreasing the incidence of bullying at work [49]. 

Supervisors design aspects of the work environment and work processes [50,51], pose demands or provide resources [52,53], act as role models for their employees [54,55,56], and directly interact with their employees through their leadership behavior and leadership style [57,58,59]. Previous studies have shown that positive leadership behaviors and styles, such as appreciation, supervisor support, and transformational leadership, are particularly beneficial to the employees’ health. In contrast, negative leadership behaviors, such as supervisor abuse, can lead to a lasting impairment of employees’ health [19,60,61,62,63,64,65].

A meta-analysis of publications in the clinical field [66] identified nine structures or leadership practices essential to promote HWE (healthy workplace environments). These include quality leadership at all levels in the organization; availability of and support for education, career, performance, and competence development; administrative sanction for autonomous and collaborative practice; evidence-based practice education and operational supports; culture, practice, and opportunity to learn interdisciplinary collaboration; empowered, shared decision-making structures and; development and support of intradisciplinary teamwork staff. Leadership’s day-to-day fluctuations have also been demonstrated to have a meaningful impact on employees’ engagement and motivation [50]. Therefore, it is paramount to raise leaders’ awareness of their behavior by providing them with training and feedback from their subordinates.

Regarding the effect of organizational culture on job satisfaction, the literature suggests differences in what values women and men see as essential to be satisfied with their jobs [67]. For example, females tend to value attention to detail, equal opportunities, workplace relationships, and balance between work and personal life. On the other hand, men disregard attention to detail, stability, and informality and appreciate ambition, dominance, and transparent rules. Organizational cultures that adopt a more humane orientation, gender equity practices, and low power distance have proved to promote more women’s professional progress [68]. Feminine leadership has also demonstrated to contribute to non-aggressive and non-competitive organizational cultures, facilitating teamwork and collaboration [69].

Based on the hypothesis that differences in business leadership exercised by men and women can influence the well-being of workers, the main objective of this study is to show a dual perspective of how all employees of these sampled companies understand and perceive the differences between male and female business leaders and how their managers exercise this leadership in order to determine that the work environment generated by these leadership differences in the company has a direct impact on their health. 

## 2. Materials and Methods

Participants. The data was collected in the Basque Country, a prosperous region in the north of Spain. In innovation, the Basque Country exceeds the EU average. Specifically, it is at 103.6% over the EU-27 and in 93rd position out of 240 European regions (https://innovacioneuskadi.innobasque.eus/euskadi-en-ue, accessed on 19 December 2021). The total number of participants was 549 employees from 16 companies. On average, the sample participants were 41.5 years old, with a mean service of 11.4 years in the company. They belong to various sectors, including new technologies (120), industry (140), construction (64), commerce and hospitality (71), education (41), and banking (113). Regarding the size of the companies, 5 are small companies (between 50–100 employees), 4 are medium-sized companies (101–250 employees), 8 are large companies (+250 employees).

A total of 277 (50.5%) of the participants were men, and 272 (49.5%) were women. According to position and responsibility in the company, the sample comprises 63 managers, 170 middle managers, and 316 employees. In our work, we have focused on managers (not middle managers), since they represent the highest level of leadership in the company. Regarding years of seniority in the company, 97 employees have less than 2 years of seniority, 84 between 3–5 years, 52 between 6–10 years, 154 between 11–20 years, and 162 more than 20 years of seniority in the company. We find a range of employees in which 62.1% are over 40 years of age.

### 2.1. Measuring Instrument

An ad hoc questionnaire was conducted, with personal information form, in addition to the Multifactor Leadership Questionnaire MLQ-6 S [70,71]. The MLQ was developed by Bass and Avolio based on research across multiple disciplines, showing excellent results for its construct validity dimensions and weighing ac-curacy (reliability) [72]. The MLQ-6S measures three different leadership styles: Transformational Leadership, Transactional Leadership, and passive/avoidant (Laissez-Faire) leadership, under seven dimensions. It uses a 5-point Likert scale (0 = Not at all to 4 = Frequently, but not always). The scores of the three items are combined to be categorized as High (9–12), Moderate (5–8) and Low (1–4). 

### 2.2. Procedure

To carry out the empirical study, the participation of each of the 16 selected companies was requested by telephone (to CEO or President). After acceptance, a message was sent to respective Human Resources Director including a link to Qualtrics software platform, through which employees could take part in the study [73]. As soon as the company agreed to participate (after giving its approval to the content of the questionnaire), the staff and managers had two weeks to respond to the questionnaire. The survey asked about key aspects of management styles and employee perceptions and was completed by 549 participants (managers and employees) between October and November 2020. 

The confidentiality of the data and the privacy of both the participants and the companies have been guaranteed.

### 2.3. Research Method

The methodology used in this study is quantitative descriptive [74] since the purpose of the research is to describe the context of the work environment in these companies, according to the differences in male or female leadership, which in turn, affects the well-being of employees.

On the one hand, we will analyze how such context affects employees based on their years of experience, the sector of the companies in which they work, their age, seniority in the company, current position in the company, whether their manager is a man or a woman, the qualities that they value in a good leader and the difficulties that women currently face when exercising leadership in companies. On the other hand, we will focus on how managers believe they exercise leadership and the guidelines they follow to do so.

With all this data, we will obtain a model that will help us understand whether the differences in male and female leadership in companies describe a more or less favorable scenario for employees to feel comfortable, well treated, and directed when performing their work. Considering that previous works indicate that a good work environment promotes employees’ health and well-being and that managers have an important role in how such environments are built, distinguishing differences in how employees perceive leadership practices carried out by men and women can support companies in implementing initiatives accordingly in order to create healthy work environments.

## 3. Results and Discussion

To carry out the analysis of our research, we will first present how the 549 respondents for this research value the differences in business leadership between men and women in their businesses.

### 3.1. How Is Good Business Leadership and Its Gender Perspective

#### 3.1.1. Qualities of a Good Leader at Work

When it comes to exercising leadership, different qualities are required to be an efficient leader that not everyone possesses and that have made a difference when the leader is either a man or a woman. Among these characteristics, we have considered the following features, differentiating between what the total of Men (M = 50.46%) and the total of Women (W = 49.54%) think.

Women’s capabilities are increasingly recognized as complementary to business goals [75]—skills developed particularly well from the house responsibilities, such as multitasking, interpersonal skills, and emotional empathy [76]. Other research from Lucas [77] shows that as more women are seen leading business, the notion becomes institutionalized and less foreign. This phenomenon helps to shift cultures in a way that reduces barriers to women by adding legitimacy, so that the notion of women in leadership becomes increasingly engrained in a culture’s social and economic fabric [75]. In reality, however, there may not be so many differences when it comes to exercising good leadership, and this may have an impact on the health of workers.

When it comes to maintaining a tone of respect (L) that facilitates good communication between bosses and employees, it seems that there is usually not a great difference in treatment. Although, as shown in Figure 1, there is a tendency to think that women are somewhat better (M = 8.38%, W = 9.29%) or much better (M = 3.28%, W = 3.64%) than men in this regard.

As to defending what they believe (M), men consider to a greater extent (M = 40.62%, W = 36.98%) that there are no great differences with women. Furthermore, women seem to think that they do somewhat better (M = 4.37%, W = 6.38%) or much better than men (M = 1.28%, W = 3.10%). This could be because it may be harder for them to demonstrate what they do or propose and they have to defend it more.

Something similar happens when it comes to being persuasive as leaders (N). With higher percentages than in the previous cases, women and men think women are somewhat more persuasive than men (M = 8.50%, W = 10.38%).

It seems clear so far that if women communicate better with their subordinates, they are more able to convince them by defending their beliefs about how to handle things at work.

Regarding who works best under pressure (O), there are no further differences. Women perceive themselves positively since they consider that there are no differences W = 33.33%, they do somewhat better than men W = 7.83% or much better W = 3.28%. Men mostly believe that there are no differences from women when working under pressure (M = 37.89%) or that they are the ones who do somewhat better (M = 6.19%). It seems that something is changing in recent times and that men perceive the new leadership exercised by women in a more balanced way. 

Taking risks (P) seems not to be characteristic of women’s leadership and what men and women seem to agree on (M = 11.29%, W = 10.38%, men exercise it somewhat better). This, which is, in fact, a particular characteristic of leadership, should be something that girls could work on as leaders of the future, which is currently a pending subject to be dealt with from school.

In the management of mixed teams (Q), the propensity of women to think that they do somewhat better (W = 9.65%) or much better (W = 3.10%) than men continues. For their part, men continue to think to a great extent that there are no differences in managing teams (M = 38.07%) and even that they do it somewhat better (M = 5.10%). This data confirms that men are more predisposed than women to think that there are no differences in exercising leadership between men and women; and that women are, in any case, the ones who believe they can do somewhat better or much better than men.

As for how the different capacities of people are valued (R) and how team members are accompanied (S), men do not perceive that there are differences compared to how women perform (M = 40.62%). This confirms, once more, women’s inclination to believe that they do somewhat better or much better than men (W = 12.75 and W = 12.57%). These two characteristics of leadership go hand in hand when it comes to creating a positive work environment. Being able to value the things that employees do well and being with them in the process is essential for workers to feel comfortable and recognized. Therefore, this positively affects their health.

If we add to this that negotiating beneficial contracts for the company (T) is where both men (M = 42.44%) and women (W = 41.53%) agree that there are no differences, which is in itself a great advance, it seems that the qualities that define a good leader are increasingly balanced in terms of gender.L. Maintaining the tone of respect in the work environment;M. Defending what they believe;N. Being persuasive;O. Working under pressure;P. Willingness to take risks;Q. Managing mixed teams (men and women);R. Valuing the different capacities of people;S. Accompanying people who are part of the teams;T. Negotiating beneficial contracts for the company.

**Figure 1 ijerph-19-00030-f001:**
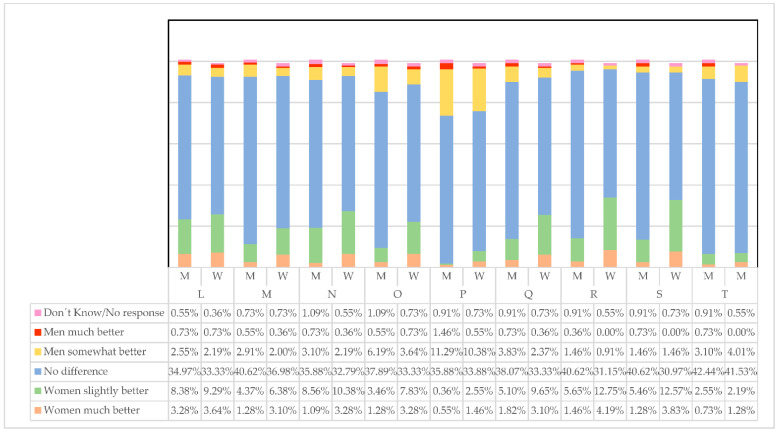
Qualities required to be an efficient leader. Source: Own elaboration.

#### 3.1.2. Women in Management Positions

As we have seen so far, the tendency regarding the different characteristics of leadership is to think that, in general, there are no significant differences when it comes to exercising it and that, in any case, women consider they do things somewhat better.

For the respondents (Figure 2), men and women lead differently (27.50% somewhat agree, 19.67% fully agree), which is understandable considering the different gender roles they have exercised historically. The question now is to study if these differences are beneficial for the companies’ results, the work environment, and the consequent health of the workers.

There is the idea that there should be more women in managerial positions than there are now 37.52% (Figure 3), or in any case, more women than men (2.19%). However, it is striking that for 30.05%, it is an indifferent question or that for 7.10%, there are already too many women in managerial positions when there is evidence that fewer women are occupying these positions.

#### 3.1.3. Perception of Female Business Leadership

Below we present different concepts of how leadership exercised by women is observed at the business level and the way they have to reach it. Regarding whether family responsibilities make it more difficult for women to reach these managerial positions (V), a large percentage of women agree (W = 24.95%) or somewhat agree (W = 16.76%) (Figure 4), while men are to a lesser extent, somewhat in agreement (M = 22.95%). Women have been dragging these family responsibilities from the past, and thus, it is more difficult for them to reconcile with management positions. Less agreement with men is perceived in this regard.

For women, they have to do more than men to prove their worth (W), with which they completely agree (W = 22.77%) or somewhat agree (W = 14.39%). For their part, men only somewhat agree (M = 15.30%) or rather, completely disagree (M = 11.48%). This discrepancy in criteria may be because men have been practicing managers for many years and have had easier and more direct access to these positions without being very aware of the difficulties women have for accessing them.

There is less support from male leaders towards women (X), with which women themselves completely agree (M = 13.11%) or somewhat agree (M = 18.76%). At the same time, the perception of men remains polarized between those who think that they somewhat agree (M = 15.66%) or completely disagree (M = 10.20%). These differences could have their origin in the fact that men consider that in general, it is not more difficult for women to become leaders.

From a women’s point of view, many men are not prepared to have a female manager (Y) (W = 21.49% somewhat agree, W = 14.21% completely agree). In this regard, men are more lukewarm when speaking (M = 16.58% somewhat agree, M = 12.57% neither agree nor disagree).

The difficulties exposed so far mean that not all women are interested in being managers or directors (Z), in which there seems to be a more significant agreement. On the other hand, most respondents strongly disagree that women are not tough enough to be managers (AA).V. Family responsibilities make it more difficult for women to reach managerial positions;W. Women have to do more than men to prove their worth;X. Women receive less support from male leaders;Y. Many men are not prepared to have a female manager or boss;Z. Not all women have an interest in being managers or directors;AA. Women are not tough enough to be managers or bosses.

**Figure 4 ijerph-19-00030-f004:**
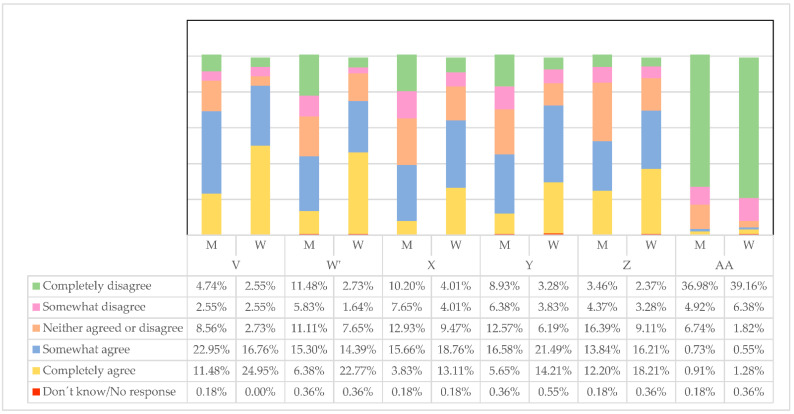
Perception of female business leadership. Source: Own elaboration.

### 3.2. Leadership Style Exercised by Managers

It is well known that the workplace, whether public or private, can have a significant influence on injuries, illnesses, deaths and threats to environmental, community and individual well-being, and this influence can be positive or negative [78].

In this section, we want to make visible the leadership qualities exercised by the 63 managers, identified as such, in the companies analyzed. Among them were 38 men and 25 women, which a priori already informs us of the lower accessibility of women to managerial positions. We will explore their answers and review the considerations of men in contrast to those of women.

The first reflection that we request of the managers, about how they think they make those around them feel (AD), is undoubtedly of special importance for this research. If a manager exercises his leadership well, they should do so assuming that the treatment of their employees is strict but cordial. Only in this way will the work environment be conducive to achieving better results for the company and the health of the workers. Both men and women think in similar percentages that quite often (M = 42.11%, W = 48.00%) or frequently but not always (M = 36.84%, W = 36.00%) managers make them feel good (Figure 5). This means that both groups believe they are doing it appropriately.

Following the trend we mentioned earlier about the better communication of women, women as managers consider they can express what employees could and should do in a simpler way than men (W = 52.00% quite often, W = 68.42% frequently, but not always). Although men also have a good concept of themselves in this regard.

Regarding the help that leaders offer their employees to think with new approaches (AF), it is women who seem to provide new ideas (W = 64.00%) much more often than men (M = 36.84%). This makes sense since the greater incorporation of women into managerial positions supposes new atmospheres and ways of doing things.

Regarding the help that men offer their subordinates to develop (AG), they believe they have more opportunities than women to do so frequently, but not always (M = 36.84%, W = 24.00%). This may be due to a greater network of contacts, years of experience, and sector knowledge.

Female managers tend to tell others to a greater extent what they have to do if they want to be well valued in their work (AH). This may be due to the greater difficulties they have had when it comes to becoming promoted and being recognized, so they tend to detect and comment on what employees must show to be well seen. Some (W = 24.00%) women do it frequently but not always, while on the contrary, men don’t do it at all (M = 7.89%).

Men, frequently but not always, seem to be more satisfied when others do their job well (AI) (M = 71.05%) compared to (W = 52%) women. This may be because men have more tradition as managers, years of experience, and another way of directing, making them more aware of their subordinates, while women focus on other objectives.

Regarding letting employees do their work as they had been doing (AJ), men are somewhat more likely (M = 47.37%) than women (W = 40%). In any case, women are unmarked by giving their opinion (W = 12%) not agreeing at all, compared to (M = 5.26%) men. This may mean that women want to change the ways of working and develop new ideas different from those historically established by men.AD. I make those around me feel good;AE. I express in a simple way what we could and should do;AF. I help to think about problems with new approaches;AG. I help others to develop;AH. I tell others what to do if they want to be valued in their work;AI. I am satisfied when others do their job well;AJ. I prefer to let others do the work as they have always done.

**Figure 5 ijerph-19-00030-f005:**
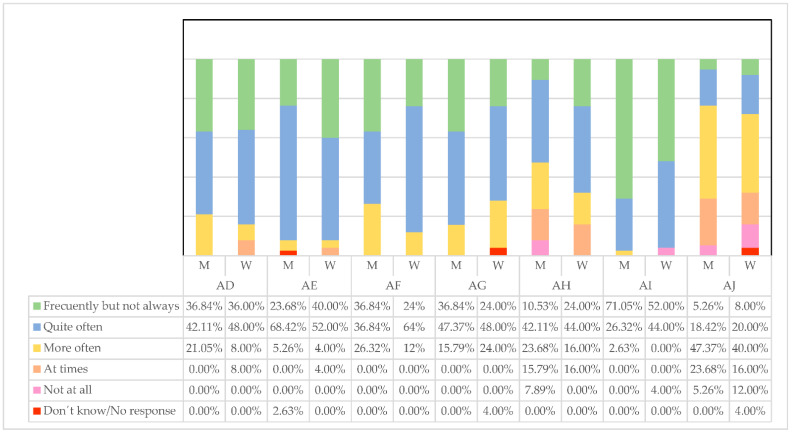
Qualities developed by managers. Source: Own elaboration.

That other employees fully trust managers (AK) is a sign of good leadership. In this sense, there are no significant differences between what men and women think, but women managers do slightly believe that employees trust them somewhat less than men. In fact, 4.00% of women do not know or do not answer (Figure 6). This lack of self-confidence seems to be decreasing in women, but similar to many other leadership characteristics, it repeats canons.

Knowing how to incentivize employees (AL) is also a sign of good leadership. Men are clearer and say they do it (M = 23.68%) more frequently than women (W = 16.00%). Men also provide others with new ways of facing challenges (MA), quite often (M = 47.37%) and frequently but not always (M = 26.32%). In this work process, female managers most inform others about how they think they are performing (AN), (W = 68%, M = 50% quite often), which highlights their good communication skills.

Rewarding people on the team with recognition or a reward when they reach their objectives (AO) is a little more common among male managers. These old techniques can create discomfort by rewarding individually rather than collectively.

“As long as things work out better, do not change anything” (AP) is a motto with which female executives do not precisely agree at all (M = 12.00%) or very occasionally (M = 40.00%). Women leaders are committed to changes and tend to be fine with what others want to do (AQ) at times (W = 40.00%, M = 28.95%), which arises from them freely.AK. Others fully trust me;TL. I know how to make the challenges we have attractive;AM. I provide others with new ways to face challenges;AN. I inform others about how I think they are performing;AO. I provide recognition/reward to team members when they reach their goals;AP. As long as things are working, it is better not to change anything;AQ. Everything that others want to do is fine with me.

**Figure 6 ijerph-19-00030-f006:**
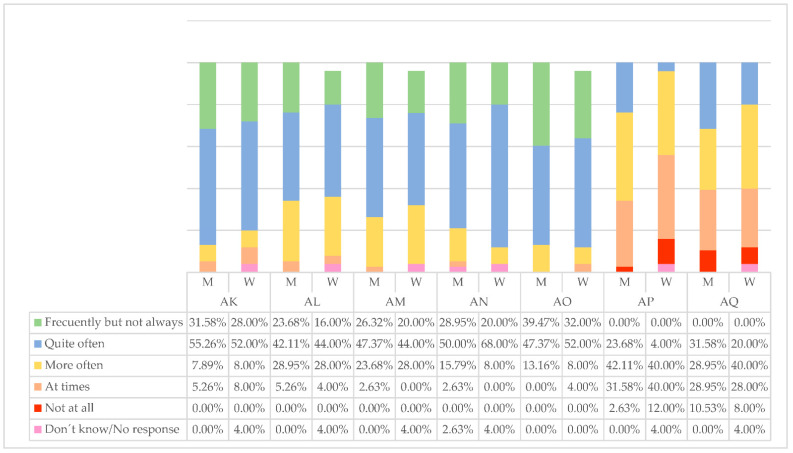
Qualities developed by managers. Source: own elaboration.

Regarding the positive image that managers believe to have, which makes people feel proud of being related to them (AR), we can say that, by far, men are quite often (M = 60.53%) very sure of it (Figure 7). On the other hand, women doubt more often about that good concept that others may have of them. It may be a matter of insecurities that are still in force due to the smaller number of women managers, which will hopefully improve over the years.

Regarding the eagerness to help, in this case for others to find meaning in their work (SA), men who exercise leadership seem to be somewhat more likely (M = 65.79%) than women (W = 60%) to lend a hand to others. At the same time, men make others rethink ideas that have never been questioned before (AT) quite often (M = 50.00%) compared to women (W = 32%). Men believe they give more personal attention to distressed employees (UA) than women, while women doubt that they are doing this well.

In terms of the public recognition of employee achievements (AV), there are few differences, but we can say that some women have certain doubts about whether they recognize these merits.

Following the precept that women communicate better, female managers tend to speak about the guidelines or standards that must be known to carry out the work (AW) and make it more productive.

Finally, men and women believe that they demand of others only what is strictly necessary, in a similar way. Both believe they create more relaxed and untroubled environments.
AR. People are proud to be related to me;ACE. I help others to find meaning in their work;AT. I get others to rethink ideas that have never been questioned before;AU. I give personal attention to employees with difficulties;AV. I make public recognition of the achievements of others;AW. I tell others the guidelines/standards they must know to carry out their work;AX. I only demand from others what is strictly necessary.

**Figure 7 ijerph-19-00030-f007:**
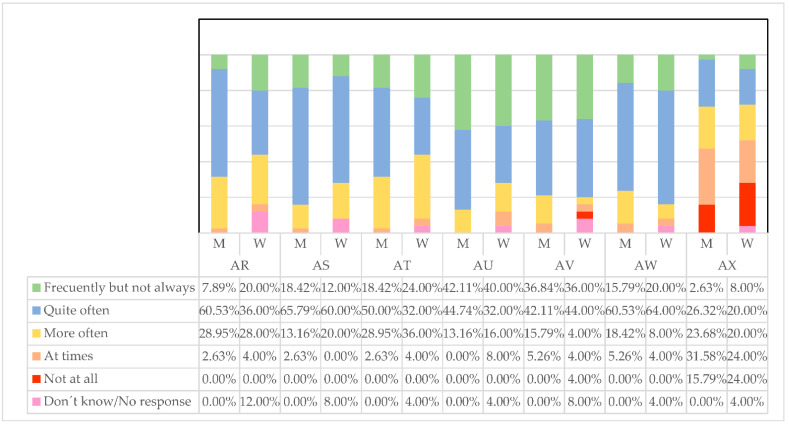
Qualities developed by managers. Source: own elaboration.

## 4. Conclusions

It is known that leadership behaviors can have spillover effects beyond the work boundary [79]. The concept of health-promoting leadership has emerged as a salient factor: it not only expands the traditional leadership theory, but also has practical implications for confronting the tricky issues of employees’ health [80]. Health-promoting leadership is positively related to employee job satisfaction, job engagement, and affective commitment [81], and negatively related to job burnout and interpersonal conflict [82,83].

In this case, the research results show us that there are no perceived differences in the qualities of the leader between men and women, although there are some tendencies that we are going to expose below.

We can say that today women have an excellent concept of themselves when they act as managers. For men, there are no significant differences when it comes to exercising leadership compared to women. In this sense, there is a tendency, on the part of employees, to think that women communicate better with their subordinates, in a respectful way. They are also better able to convince them by defending their beliefs of how to make things run at work. In turn, women are more likely to value the things their subordinates do well and be with them in the process, which is essential for employees to feel comfortable and recognized, affecting their health positively. As a pending issue, women, although they negotiate well, improve their predisposition to take risks to, perhaps, improve business results.

In general, men and women do not lead differently, but there are some non-significant nuances or trends derived from the different gender roles that have been exercised historically. Now we should ask ourselves if these differences are beneficial for the companies’ results and improve the work environment and the consequent health of the workers.

Regarding the perception of female leadership and how it is reached, the men who participated in this study consider that, in general, it is not more difficult for women to become leaders. On the contrary, women believe they have to face a series of conditions that make it challenging to be leaders in companies. They have more family responsibilities. They believe they must do more than men to prove themselves. They feel less support from male leaders. In addition, many men are not prepared to have a woman as a manager, since so far, it has been somewhat less frequent. All this means that not all women are interested in being managers, although the vast majority of those surveyed strongly disagree that women are not tough enough to be so. In this regard, there is a clear need to raise male managers’ awareness of the difficulties women encounter when climbing up organizational stairs. In this way, male managers might become more prone to adopt more humane and gender equality measures to promote women’s progression in the workplace [68].

According to managers, when it comes to exercising leadership, both men and women have similar opinions that quite often (M = 42.11%, W = 48.00%) or frequently, but not always (M = 36.84%, W = 36.00%), they make their employees feel good. This means that both groups believe they are performing appropriately. Likewise, women managers consider expressing what workers could and should do more simply and efficiently than men. They are willing to offer more help to their employees to think with new approaches, adding new habits that change the routines established by male leadership in a friendlier environment. However, the support offered by men is more linked to the professional development of their employees. This may be due to a larger network of contacts, years of experience, and sector knowledge.

Having a quiet climate at work can play an essential role in men’s willingness and ability to disrupt sexist behavior. When those with power in organizations do not encourage people to speak up, employees are more likely to keep quiet when they witness sexism against women [84].

Women in leadership positions tend to tell others, to a greater extent, what to do if they want to be valued in their work. This may be a consequence of the greater difficulties that women have had when it comes to ascending and being recognized. On the other hand, men don’t do so (M = 7.89%). This coincides with previous literature that has also shown that women tend to evaluate both attention to detail and having equal opportunities in the workplace [67]. As stated by some women leaders interviewed in the KPMG report [85] “having had a more formal training in effective leadership following these types of guidelines would have made women less reluctant to assume these functions, since it is difficult for us to fight for those opportunities”.

Furthermore, male managers feel more satisfied than women when employees fulfill their duties, allowing them to continue doing this as they had been doing. This perpetuates gender roles. Instead, women seek to change the ways of working, contributing new ideas, different from those historically established. This can change the work environment in companies and, with good communication and treatment, to which women seem to be willing, it can contribute to better health and well-being of employees. Those work environments with perpetuated bad work climates have the opportunity to leave the vicious circle by changing the leadership style. In this sense, female leadership can be a valuable opportunity for companies.

On the part of female managers, there is a slight tendency to believe that employees trust them somewhat less than in men. This lack of self-confidence seems to be decreasing, but similar to many other perceptions, it repeats canons.

On the other hand, male managers believe that they know how to incentivize their employees better, providing them with new ways to face challenges and rewarding achievements.

In this day-to-day, female managers are the ones who most inform others about how they think they are performing. They use these techniques partly because they are very much in favor of the dialogue promoting more knowledge, tranquility, and profound changes. As Kawatra and Krishnan [69] argue, a feminine leadership style that tends towards dialogue and relationships can better contribute to non-aggressive and non-competitive organizational culture.

Male managers believe that they help more to make sense of the work that their subordinates do while also rethinking ideas that have never been questioned before. Women leaders have a worse concept of themselves on these last points. They even have doubts about the perception that others may have of them.

From the perspective of male managers, they believe that they help others find meaning in their work, rethink ideas that have never been questioned before or give more personal attention to employees with difficulties. In contrast, female managers have more doubts about doing this right. They are also not particularly clear if they publicly acknowledge the merits of others, but they do communicate better. Men and women demand of others only what they believe is strictly necessary, similarly. They think they create more relaxed and untroubled environments.

In summary, it is necessary to differentiate, on the one hand, how employees perceive that men and women exercise leadership, and on the other, what men and women as managers think about how they carry out this work. In this sense, there is a contradiction that may be hindering women themselves. From the outside, they have a good concept of themselves, although in practice and as managers, they want to change things and still have some insecurities. Men, for their part, do not believe that there are great differences when it comes to exercising leadership compared to women. As managers, they are more confident that they do their job well.

Communication is the tool that women managers know best how to work with and that some men, in their turn, still need to develop. The greatest successes can be achieved for the company and the health and well-being of the employees by using the right words, in a relaxed tone of dialogue, in a way that explains, motivates, and serves to bring about changes in the work environment. It is undoubtedly a field to be analyzed in future research.

## Figures and Tables

**Figure 2 ijerph-19-00030-f002:**
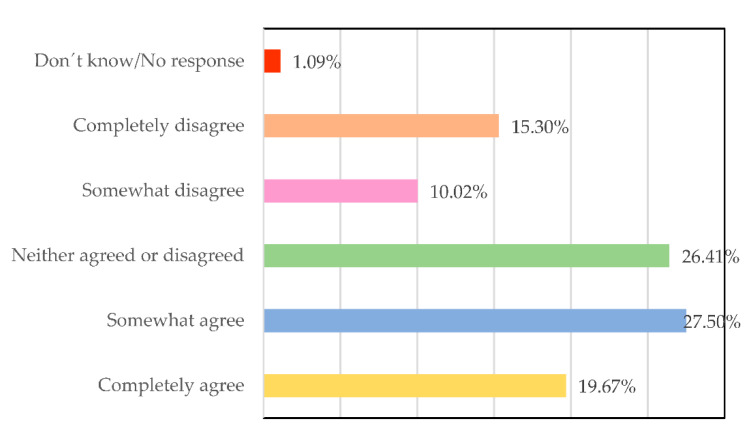
Women and men lead differently. Source: Own elaboration.

**Figure 3 ijerph-19-00030-f003:**
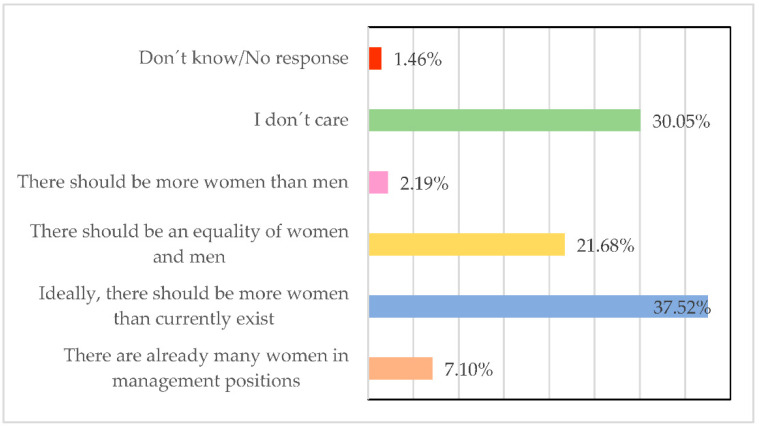
Number of women in managerial positions in your company. Source: Own elaboration.

## Data Availability

The data presented in this study are available upon request from the corresponding author. The data are not publicly available because they are not available on a public domain server, but are in the private research repository of the University of Deusto.

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
