# Peer review of "Business Leadership from a Gender Perspective and Its Impact on the Work Environment and Employee’s Well-Being in Companies in the Basque Country"

_ijerph, 2021, doi:10.3390/ijerph19010030_

Round 1

Reviewer 1 Report

Business leadership from a gender perspective and its impact on the work environment and employee's well-being.”

 I believe this article should be published as is, with the sole exception of having an English native speaker revise it one last time.  Some sentences sound awkward, perhaps due to having been translated from another language? 

I see no major concerns in relation to the issues discussed in the essay, and I really welcome the gender perspective informing the research. 

However, I would like to make a few suggestions to the author.  Perhaps they could deem them worth exploring or to be included in their future research:

  1. There are a few typos, a few places where a sentence seems to be missing, or moments in the paper where it is difficult to follow. Please have someone edit the paper since the dissemination of its research findings will benefit from some minor editing. 

  1. An extra effort could be made in correlating the workers’ health and well-being concepts with the questions posed to the employees in those 16 companies in the Basque Country in Spain. How are those questions conducive to a healthy work environment?  In what ways?  Perhaps point 1.1 could be developed further in future articles and research.  It is not always clear why and how perceived impressions about leadership—the survey conducted—help enhance or decrease a good working environment.  As a matter of fact, well-being and health seem to have disappeared in the conclusion.

  1. I was curious about the specific working conditions in companies in the Basque Country. Is there anything specific about them?  Is the legislation different from other Western European countries?  Is it different from Spain?    

  1. How could more factors be incorporated in the survey elaborated by the authors? For example, could questions related to a welfare state (more vacation time, more days to take care of personal affairs, of an elderly parent, maternity leave legislation, well-established processes to resolve labor or personal disputes, etc.) be incorporated in the survey?  How could these collective and public policy issues affect the perceived impressions about business leadership?

  1. On page 13, the author says, “there is a clear need to raise male managers’ awareness of the difficulties women encounter when climbing up organizational status. By doing so, male managers might be more prone to adopt [. . .] gender equality measures.”  I wonder if public policy measures regarding gender equality will be more effective than depending on the sole willingness of male or female managers to implement them.  Could this issue be incorporated in further surveys?

  1. Perphaps it should be made clear in the title that the research referes to companies in the Basque Country. Only a suggestion. 

I hope the author finds my comments useful.  It has been a pleasure to read their work. 

Author Response

Dear Editors,

We thank you very much for the suggestions and contributions you have made to our research. Below are the changes we have made.

Cordially

Reviewer 2 Report

The article is of interest and topicality as it addresses business leadership from a gender perspective and the emotional well-being of employees. However, the following considerations should be taken into account in the structure of the article in order to improve it.

The objective(s) of the research should be reflected at the end of the Introduction section and it is also advisable to establish the starting hypotheses, just before section 2.

Secondly, section 2 Materials and methods should be structured as follows:

2.1. Participants, including all information concerning the participants, mean age, standard deviation, distribution by economic sectors/companies....

2.2. Measuring instruments: describing the instrument used to collect information, as well as its factors and dimensions and its psychometric properties (reliability and validity).

2.3. Procedure. Describing the information collection process (Qualtric software), the time in which it was applied, the request for permission from the companies and personnel to whom it was administered, etc.

2.4. Research method: describing the phases of collection, analysis of the information and type of analysis carried out.

  1. Results and discussion

In the discussion section, the authors should take up the degree of fulfillment of both the objectives and the initial hypotheses, contrasting them with the results obtained in other investigations, so that the reader can understand whether or not the results are conclusive.

  1. Conclusions

It is advisable to include more quotations from the authors of the conclusions that are being drawn.

References

It is convenient that they are in APA 7th edition format and that they correct errors such as those that come in the quotation nº 73 in which they have put the link to their own hard disk

  1. KPMG (2015). "Women's Leadership Study. Moving Woman Forward into Leadership Roles." file:///C:/Users/34722/Downloads/final_womens_leadership_v19.pdf.

Author Response

(The authors gave the same response as above.)
